# COVID-19, Non-Communicable Diseases, and Behavioral Factors in the Peruvian Population ≥ 15 Years: An Ecological Study during the First and Second Year of the COVID-19 Pandemic

**DOI:** 10.3390/ijerph191811757

**Published:** 2022-09-17

**Authors:** Jordan Canorio, Flor Sánchez, Max Carlos Ramírez-Soto

**Affiliations:** Facultad de Ciencias de la Salud, Universidad Tecnológica del Peru, Lima 15046, Peru

**Keywords:** COVID-19, obesity, alcohol use prevalence, mortality, incidence, case fatality rate

## Abstract

A range of health-related and behavioral risk factors are associated with COVID-19 incidence and mortality. In the present study, we assess the association between incidence, mortality, and case fatality rate due to COVID-19 and the prevalence of hypertension, obesity, overweight, tobacco and alcohol use in the Peruvian population aged ≥15 years during the first and second year of the COVID-19 pandemic. In this ecological study, we used the prevalence rates of hypertension, overweight, obesity, tobacco, and alcohol use obtained from the Encuesta Demográfica y de Salud Familiar (ENDES) 2020 and 2021. We estimated the crude incidence and mortality rates (per 100,000 habitants) and case fatality rate (%) of COVID-19 in 25 Peruvian regions using data from the Peruvian Ministry of Health that were accurate as of 31 December 2021. Spearman correlation and lineal regression analysis was applied to assess the correlations between the study variables as well as multivariable regression analysis adjusted by confounding factors affecting the incidence and mortality rate and case fatality rate of COVID-19. In 2020, adjusted by confounding factors, the prevalence rate of obesity (β = 0.582; *p* = 0.037) was found to be associated with the COVID-19 mortality rate (per 100,000 habitants). There was also an association between obesity and the COVID-19 case fatality rate (β = 0.993; *p* = 0.014). In 2021, the prevalence of obesity was also found to be associated with the COVID-19 mortality rate (β = 0.713; *p* = 0.028); however, adjusted by confounding factors, including COVID-19 vaccination coverage rates, no association was found between the obesity prevalence and the COVID-19 mortality rate (β = 0.031; *p* = 0.895). In summary, Peruvian regions with higher obesity prevalence rates had higher COVID-19 mortality and case fatality rates during the first year of the COVID-19 pandemic. However, adjusted by the COVID-19 vaccination coverage, no association between the obesity prevalence rate and the COVID-19 mortality rate was found during the second year of the COVID-19 pandemic.

## 1. Introduction

The incidence and mortality due to COVID-19 is associated with multiple risk factors, such as sociodemographic and health indexes, and genetic and immunology factors [1]. Among these, hypertension and obesity are significant risk factors for severe illness and death [2,3,4,5]. A recent study suggested that countries with a higher body mass index (BMI) or cholesterol level have a higher rate of COVID-19 incidence and mortality [6]. In addition, obesity is associated with the risk of hospitalization and mortality due to COVID-19 [3,5]. Behavioral risk factors have also been found to be associated with COVID-19 outcomes, above all in some regions where the prevalence of tobacco and alcohol use is high [7]. A systematic review suggested that smokers appear to be at a low risk of SARS-CoV-2 infection, while former smokers appear to be at increased risk of hospitalization, increased disease severity, and mortality from COVID-19 [8]. Another study examined whether social and behavioral risk factors were associated with mortality among U.S. veterans with COVID-19, but no association was found [9]. Despite these findings, we remain unclear about whether behavioral risk factors are associated with incidence and mortality due to COVID-19.

Peru has been deeply affected the COVID-19 pandemic. By the end of December 2021, more than 4 million confirmed COVID-19 cases and more than 215,000 deaths had been reported [10]. In addition, recent studies revealed that more than 100,000 excess all-cause deaths occurred in 2020, and there was a strong excess in geographical and temporal mortality patterns according to region [11]. This excess mortality can be explained by social factors, individual factors, the health care system, and other direct and indirect factors related to COVID-19 infection. Several Peruvian studies have shown an association between obesity and COVID-19 mortality, both in the overall population and in patients infected with SARS-CoV-2 [12,13]. To control the COVID-19 pandemic and mitigate mortality associated with its complications, Peru launched the COVID-19 vaccination on 9 February 2021, which initially included immunization of health workers and subsequently encompassed the overall population [14]. During the post-vaccination months, there has been a reduction in the COVID-19 mortality rate in the Peruvian population [15,16]. In this setting, we assessed the association between incidence, mortality, and case fatality rate due to COVID-19 and the prevalence rates of hypertension, obesity, overweight, tobacco, and alcohol use in the Peruvian population aged >15 years during the first and second year (since the launch of the COVID-19 vaccination) of the COVID-19 pandemic. These findings will help to determine the indirect impact of the SARS-CoV-2 vaccination on the COVID-19 mortality risk in regions where the prevalence rates of hypertension, overweight, obesity, smoking, and alcohol use are high.

## 2. Materials and Methods

### 2.1. Design and Area Study

We conducted an ecological study to analyze the association between the prevalence rates of hypertension, overweight, obesity, tobacco use, and alcohol use and the incidence rate, mortality rate, and case fatality rate due to COVID-19 in the Peruvian population between 1 March 2020 and 31 December 2021. Peru has an estimated 33 million inhabitants, according to the National Institute of Statistics and Informatics (INEI), divided into 24 geographic regions and one constitutional province, Callao [17]. The study was performed following the Strengthening the Reporting of Observational Studies in Epidemiology (Appendix A Table A1. Checklist STROBE Statement) reporting guidelines [18].

### 2.2. Risk Factors in Peru

For this study, we retrieved secondary data of each geographic region on the prevalence rates of hypertension, overweight, obesity, tobacco use, and alcohol use in Peruvian individuals aged ≥ 15 years from the Encuesta Demografica y de Salud Familiar (ENDES) 2020 and 2021 of the INEI [19,20]. ENDES 2020 included to 32,197 men and women aged ≥ 15 years from 25 Peruvian regions, from January to December 2020, and ENDES 2021 includes to 32,124 men and women aged ≥15 years [19,20]. Hypertension was defined as an elevation above the normal values of systolic blood pressure over diastolic blood pressure of 140/90 mm Hg [19]. Overweight and obesity were defined as the disproportionate deposition of fat in the body that can damage health. Tobacco and alcohol use were defined as risk factors for developing various chronic diseases that alter cardiovascular function, liver function, and renal function, among others [19].

### 2.3. COVID-19 Data Collection

COVID-19 case data were obtained from the freely accessible National Open Data Platform website, Peru (COVID-19 Open Data, https://www.datosabiertos.gob.pe/group/datos-abiertos-de-covid-19 (accessed on 20 January 2022)), accurate as of 31 December 2021. A COVID-19 confirmed case was defined as a case with a positive result for SARS-CoV-2 from an RT-PCR assay or antigen test or immunochromatographic test. In the National Open Data Platform, COVID-19 Open Data (“COVID-19 positive cases” and “COVID-19 deaths”), and data on cases of and deaths from COVID-19 include information on sex, age, region, and date of notification (cases or deaths). Other information on cases and deaths from COVID-19 is not available in the National Open Data Platform, COVID-19 Open Data. COVID-19 deaths were obtained from the Sistema Informatico Nacional de Defunciones (SINADEF) from the Peruvian Ministry of Health (MINSA), from March 2020 to December 2021 [21]. The SINADEF database records all deaths that occur in Peru. We included the deaths by COVID-19 as the underlying cause of death.

### 2.4. Confounding Factors

Confounding factors included the mean age in the region (cases or deaths), mean income and gender balance (cases or deaths), and the number of intensive care unit (ICU) beds. These confounding factors were used in a previous observational study [12]. We also used COVID-19 vaccination coverage rates in individuals ≥ 18 years in 2021 as a confounding factor. Data on the mean age in the region, mean income, and gender balance were retrieved obtained from the INEI [17]. The number of ICU beds was obtained by way of App. F500.2 from the Superintendencia Nacional de Salud, Perú (SUSALUD) [22]. COVID-19 vaccination coverage rates in 2021 were obtained from the Single National Health Information Repository (REUNIS), MINSA, Peru [23].

### 2.5. Statistical Analysis

Crude incidence and mortality rates (per 100,000 habitants) among individuals ≥ 15 years old were calculated by dividing the number of COVID-19 cases and deaths per region in 2020 and 2021 by the population of each region. We also calculated the COVID-19 case fatality rate among those aged ≥15 years in each region. Estimated populations for calculating incidence and mortality rates were obtained from the INEI, Peru. Spearman’s test correlation and linear regression models were used to estimate the associations between the incidence, mortality, and case fatality rates due to COVID-19 and the prevalence rates of hypertension, overweight, obesity, smoking, and alcohol use. Non-parametric approaches were applied for non-normally distributed data. Additionally, multivariable regression models were applied to assess the association between COVID-19 incidence, mortality, and case fatality rates and the prevalence rates of hypertension, overweight, obesity, tobacco and alcohol use. Confounding factors to multivariable regression models in 2020 and 2021 included the mean age in the region (years), mean monthly income (PEN), gender balance in deaths, and number of ICU beds in 2020 and 2021. We also used COVID-19 vaccination coverage rates in 2021 as a confounding factor. *p*-values < 0.05 were considered significant. Scatter plots were provided to indicate the correlation between the study variables. Statistical analyses were done using StataSE 17.0 Software.

### 2.6. Ethics

As this study used publicly available cases and death data, and there was no direct patient involvement.

## 3. Results

A total of 2,206,897 COVID-19 cases from 6 March 2020 to 31 December 2021 in individuals aged ≥15 years were included, of whom 970,590 were in the incidence analysis for 2020, and 1,236,307 for 2021 (Figure 1A). During this period, 198,447 deaths from COVID-19 in individuals aged ≥15 years were also included, of whom 94,225 were included in the mortality analysis for 2020, and 108,351 were included in the mortality analysis for 2021 (Figure 1B).

Table 1 describes the prevalence, incidence, mortality, and fatality rates in the Peruvian population stratified by region in 2020. The hypertension prevalence rate was higher in Callao, and the prevalence rates of overweight and tobacco use were higher in Madre de Dios. The prevalence rates of obesity and alcohol use were higher in Moquegua and Piura, respectively. The COVID-19 incidence rate was higher in Moquegua, while the mortality rates were highest in Callao, Moquegua, Ica, and Lambayeque. The fatality case rates were highest in Piura, Lambayeque, La Libertad, Ica, and Callao regions (Table 1).

In 2021, the prevalence rates of hypertension and obesity were higher in Tacna region, and the prevalence rate of overweight was higher in Pasco region. The prevalence rates of tobacco and alcohol use were higher in Madre de Dios and Moquegua regions, respectively (Table 2). The COVID-19 incidence rates were highest in Moquegua, Callao and Lima regions. The mortality rates were highest in Ica, Callao, and Lima regions; and the fatality case rates were highest in Ica, Ucayali, and Lambayeque regions (Table 2). 

### 3.1. Correlation between the Study Variables and COVID-19 Measures in 2020

There was an association between the prevalence rates of obesity (r = 0.60; *p* = 0.001), tobacco use (r = 0.52; *p* = 0.006), and alcohol use (r = 0.53; *p* = 0.005) and COVID-19 incidence rates (per 100,000 inhabitants) (Figure 2A). There was also an association between obesity (r = 0.80; *p* = 0.0001) and alcohol use prevalence (r = 0.72; *p* = 0.0001) and COVID-19 mortality rates (per 100,000 inhabitants) (Figure 2B). Finally, there was an association between the prevalence of obesity and the COVID-19 case fatality rate (r = 0.45; *p* = 0.023) (Figure 2C).

### 3.2. Regression Analysis in 2020

Table 3 shows the results of a multiple regression analysis adjusted on the association between obesity, smoking, and alcohol use prevalence and the incidence, mortality, and case fatality rate of COVID-19. Results revealed that, adjusted by confounding factors, the prevalence rate of obesity (β = 0.582; *p* = 0.037) was associated with the COVID-19 mortality rate (per 100,000 inhabitants). There was also an association between obesity and the COVID-19 case fatality rate (β = 0.993; *p* = 0.014) (Table 3).

### 3.3. Correlation between the Study Variables and COVID-19 Measures in 2021

There was no association between hypertension, obesity, smoking and alcohol use prevalence and the COVID-19 incidence rate (per 100,000 inhabitants), except for the overweight (r = 0.451; *p* = 0.023) (Figure 3A). The overweight (r = 0.454; *p* = 0.022) and obesity prevalence (r = 0.619; *p* = 0.001) were associated with the COVID-19 mortality rate (per 100,000 inhabitants) (Figure 3B). There was no association between hypertension, obesity, overweight, smoking, and alcohol use prevalence and the COVID-19 case fatality rate (Figure 3C).

### 3.4. Regression Analysis in 2021

Table 4 shows the results of multiple regression analysis adjusted on the association between obesity and the COVID-19 mortality rate (per 100,000 inhabitants). When adjusted by confounding factors, the prevalence rate of obesity was associated with COVID-19 mortality rate (per 100,000 inhabitants) (β = 0.713; *p* = 0.028). However, when adjusted by confounding factors including COVID-19 vaccination coverage rates, there was no association between the obesity prevalence and the COVID-19 mortality rate (per 100,000 inhabitants) (β = 0.031; *p* = 0.895) (Table 4).

## 4. Discussion

In this nationally based study, we examined the association between health status (prevalence of hypertension, overweight, and obesity) and behavioral factors (tobacco and alcohol use prevalence) and the COVID-19 incidence and mortality rates using an ecological design. Despite relatively high levels of hypertension, overweight and obesity, and behavioral risk factors (tobacco and alcohol use prevalence), no association with the incidence of COVID-19 was found among individuals in the Peruvian population aged ≥15 years during the first and second year of the COVID-19 pandemic. Instead, we identified obesity as being associated with the COVID-19 mortality or fatality rate, which was consistent with findings reported in other studies.

### 4.1. Potential Explanations and Implications

Initially, we found a positive correlation between the prevalence rates of obesity, tobacco and alcohol use, and the incidence rate due to COVID-19; however, after adjusting for potential confounding factors, there was no association. With regard to obesity, our findings were different from a previous study indicating an association between BMI and cholesterol and the COVID-19 incidence rate in 193 countries [6].

Similar to the previous population-level studies that suggested an association between obesity and COVID-19 mortality in several countries, including a study in Peru [6,12], we found a strong correlation between obesity and mortality and fatality due to COVID-19 in 2020 even after adjusting for confounding factors. The variability in mortality rates observed among different Peruvian regions can be partly explained by the higher obesity prevalence rate. Therefore, we observed that as the obesity prevalence rate increased, the COVID-19 mortality rate increased in the Peruvian population. Obesity is a factor for hospitalization and adverse outcomes from COVID-19 [2,3,5]. Several studies explain the association of obesity and the adverse outcomes and mortality from COVID-19. First, obesity rate is associated with a higher prevalence of hypertension and diabetes, diseases that are associated with worse outcomes from COVID-19 [5,24]. Second, the SARS-CoV-2 virus has a high affinity for angiotensin-converting enzyme 2, which is essential for host cell infection [1,24]. This enzyme is expressed in adipose tissue; therefore, its presence in adipose tissue may exacerbate the severity of SARS-CoV-2 infection and worse outcomes from COVID-19 disease [25]. Third, another potential explanation is that the higher obesity grade may be associated to a more severe COVID-19 among obese individuals [2,3,4,5]. However, these data were not available in our study. Fourth, the association between the obesity rate and COVID-19 mortality rate also could be explained by external factors such as the capacity of the Peruvian health system (lack of ICU beds, oxygen, and medicines), social factors (sex, age or population density) [26,27], and other individual factors such as multimorbidity and frailty.

During the COVID-19 pandemic, concerns were widespread that behavioral risk factors (prevalence of tobacco and alcohol use) may be associated with a severity greater and death due to COVID-19, particularly during the first year of the COVID-19 pandemic. In that setting, we found a positive correlation between alcohol use and COVID-19 mortality in 2020, however, after adjusting for confounding factors in the study population, there was correlation between alcohol use and COVID-19 mortality. Although we found no ecological studies to explain this association, a cohort study among a population of U.S. veterans examined whether behavioral risk factors (tobacco and alcohol use) were associated with COVID-19 mortality, but researchers found no association [9]. The association found in our study may be difficult to explain since we did not investigate the frequency of alcohol consumption and complications among the alcohol-consuming population. Rather, the correlation between alcohol use and COVID-19 mortality may be attributed to external factors such as the overload of the health system (infrastructure, lack of ICU beds, oxygen, and medicines) and other individual factors such as multimorbidity and frailty in the Peruvian population. 

Among all the comparisons made in 2021, only a correlation between obesity and COVID-19 mortality was found even after adjusting for confounding factors. Curiously, when adjusted by confounding factors, including COVID-19 vaccination coverage rates in 2021, there was no association between the obesity prevalence rate and the COVID-19 mortality rate. Vaccination against SARS-CoV-2 is a more effective intervention in addressing the COVID-19 pandemic [1,4,28]. Despite patients with obesity showing a reduced immune response to the influenza A (H1N1) vaccine [29], the phase 3 clinical trials of Moderna COVID-19 and Pfizer-BioNTech COVID-19 vaccines included subgroups of individuals with obesity and severe obesity and showed effective results [28,30,31]. The efficacy of the Pfizer-BioNTech vaccine in individuals with obesity from 7 days after the second dose was 95.4% (95% CI: 86.0–99.1%) in preventing COVID-19 [30]. The efficacy of the Moderna vaccine in obese individuals from 14 days after the second dose was 94.1% (95% CI: 89.3–96.8%) [31]. In addition, a study investigated the vaccine’s effectiveness against SARS-CoV-2 infection and severe outcomes among patients with immune-mediated inflammatory diseases and found two vaccine doses to be effective against SARS-CoV-2 infection and severe COVID-19 outcomes in patients with rheumatoid arthritis, ankylosing spondylitis, inflammatory diseases, and psoriasis [32]. Peru implemented the SARS-CoV-2 vaccination in February 2021, initially including immunization of health workers, the population with comorbidities, and subsequently, the overall population. The vaccination led to a reduction in the COVID-19 mortality rate in the Peruvian population [15,16]. In this setting, our findings can help to explain the indirect impact of the SARS-CoV-2 vaccination on the COVID-19 mortality risk reduction in regions where the prevalence of obesity is high.

### 4.2. Limitations

Our study has several limitations. First is the ecological design; thus, our study shows findings at the population level and should not be generalizable to the individual level. Second, we used several information sources (ENDES 2020 and 2021, INEI, MINSA, SINADEF, and SUSALUD); therefore, our findings may have in a possible bias. Third, our estimates could have been overestimated owing to unmeasured variates, such as obesity grade, multimorbidity, and frailty in the Peruvian population. Fourth, in the multivariable regression analysis adjusted in 2021, we used the COVID-19 vaccination coverage rates in individuals ≥18 years, when the study population were individuals aged ≥15 years. Despite these limitations, our study is robust and large since it included confounding factors in the analysis.

### 4.3. Implications for Public Health

Our findings have important implications for public health. First, current data from the ENDES in Peru highlight that obesity prevalence rates in individuals aged ≥15 years vary by region, from 9.6 to 35.8% in 2020, and from 10.4 to 37.4% in 2021 [19]. Therefore, these individuals are more likely to have severe disease and higher risk to death, compared with individuals who have a normal weight. Second, our results can help us to explain the high mortality rates from COVID-19 in the Peruvian population. Therefore, although we cannot guarantee that weight loss will reduce COVID-19 mortality in the Peruvian population, interventions for preventing obesity and other chronic diseases are necessary to reduce the impact of future pandemics.

## 5. Conclusions

Peruvian regions with higher prevalence rates of obesity had higher COVID-19 mortality and case fatality rates in 2020. However, adjusted by COVID-19 vaccination coverage, no association was found between the obesity prevalence and the COVID-19 mortality rate in 2021. These findings support the need for interventions and public health messaging regarding COVID-19 prevention in obese individuals aged ≥15 years. In addition, these findings can help to explain the indirect impact of SARS-CoV-2 vaccination on COVID-19, informing future vaccine strategies. 

## Figures and Tables

**Figure 1 ijerph-19-11757-f001:**
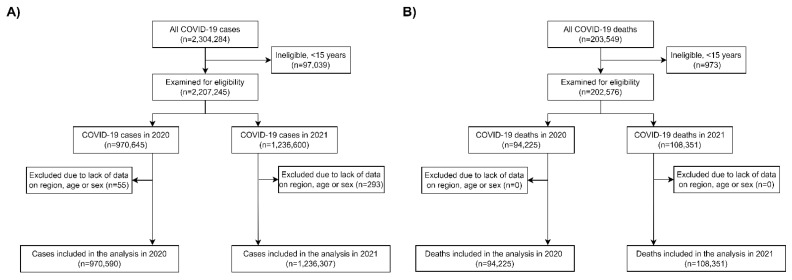
Study flow chart using the STROBE reporting guidelines. COVID-19 cases (**A**) and deaths (**B**) included in the analysis for 2020 and 2021.

**Figure 2 ijerph-19-11757-f002:**
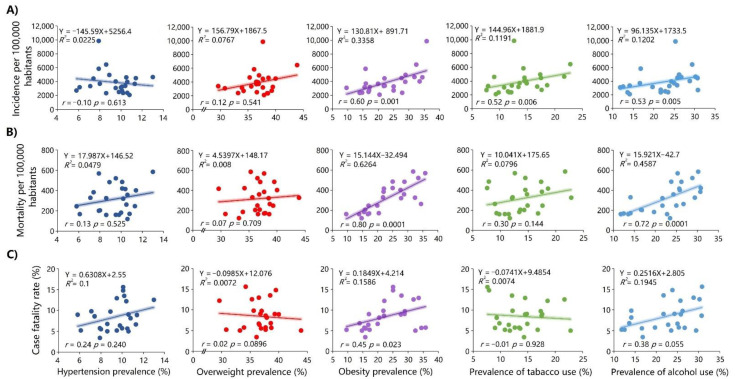
Linear regression models and Spearman correlation between the prevalence rates of hypertension, overweight, obesity, smoking, and alcohol use and the incidence rate (**A**), mortality rate (**B**), and case fatality rate due to COVID-19 (**C**) in 2020.

**Figure 3 ijerph-19-11757-f003:**
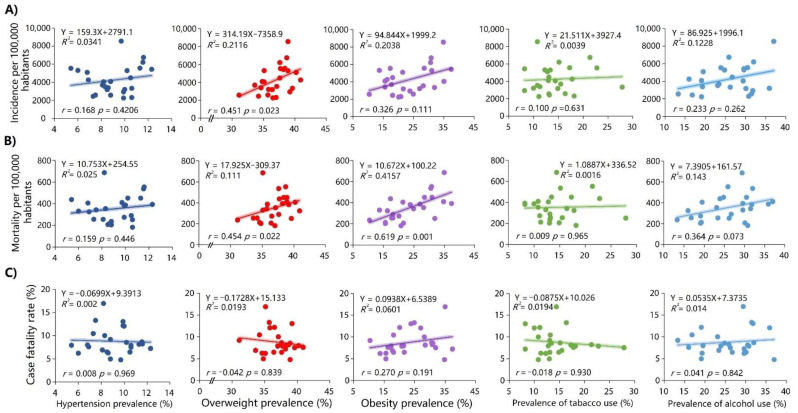
Linear regression models and Spearman correlation between the prevalence rates of hypertension, overweight, obesity, smoking, and alcohol use and the incidence rate (**A**), mortality rate (**B**), and case fatality rate due to COVID-19 (**C**) in 2021.

**Table 1 ijerph-19-11757-t001:** Prevalence rates of hypertension, overweight and obesity, smoking, and alcohol use and the incidence, mortality, and fatality rates due to COVID-19 in ≥15 year in 25 Peruvian regions in 2020.

Region	Hypertension Prevalence (%)	Overweight Prevalence (%)	Obesity Prevalence (%)	Prevalence of Tobacco Use (%)	Prevalence of Alcohol Use (%)	Population ≥ 15 Years in 2020	Incidence Rate (per 100,000 Habitants)	Mortality Rate (per 100,000 Habitants)	Case Fatality Rate (%)	Gender Balance in Deaths (Men/Women) *	Mean Age in Deaths (Years) *	Mean Monthly Income (PEN) *	No. of ICU Beds in 2020 *†
Amazonas	7.9	36.1	15.8	14.3	16.8	289,802	5852.62	201.52	3.44	2.0	64.8	992.9	14
Ancash	7.1	35.6	21.8	12.2	14.1	876,703	3351.65	328.5	9.8	2.1	68.2	1057.2	44
Apurímac	10.6	33.2	14.7	11.7	23.5	300,395	2384.53	119.84	5.03	1.5	68.1	1004.5	26
Arequipa	10.6	36.8	28.8	14.8	24.3	1,187,931	3962.01	358.1	9.04	1.2	68.9	1530.3	70
Ayacucho	10	33.9	15.5	13.4	19.7	464,136	3162.87	182.06	5.76	2.1	66.3	1095.4	20
Cajamarca	8.6	38.6	13.9	9.8	13.6	1,016,792	2348.66	159.82	6.8	2.1	68.1	850.2	26
Callao	13	35.3	31.8	21.7	30.2	902,609	4655.17	584.53	12.56	2.1	67.4	1355.6	84
Cusco	9.8	36.6	16.8	11.5	22.8	988,897	2504.61	160.89	6.42	1.3	66.0	963.1	17
Huancavelica	9.5	30.8	9.6	10.8	11.8	236,955	3083.29	161.21	5.23	2.0	66.6	669.0	10
Huánuco	8	36.9	15.9	13.6	16.6	524,371	3701.39	205.01	5.54	1.6	67.5	892.4	49
Ica	9.5	36.5	33.5	14.5	28.8	725,610	4025.72	520.67	12.93	1.7	67.1	1478.2	48
Junín	5.8	39.3	17	17.6	21.7	982,199	2728.06	245.16	8.99	1.8	65.7	1082.7	43
La Libertad	10.2	38.5	27.8	12.1	21.7	1,531,668	2366.96	319.46	13.5	1.9	67.1	1167.2	63
Lambayeque	10.1	39.5	25	7.9	23.8	991,121	3280.23	484.7	14.78	1.7	67.1	1159.6	39
Lima	8.9	37	28.9	17.35	26.2	8,750,417	4967.75	482.94	9.72	1.8	66.9	1653.5	739
Loreto	10.4	29.6	22.1	18.6	25.7	680,927	3396.69	415.02	12.22	2.0	65.6	1180.4	8
Madre de Dios	8.6	43.9	32.4	22.8	29.4	135,428	6460.26	324.9	5.03	2.8	63.0	1399.9	8
Moquegua	7.9	37.7	35.8	12.6	25.3	155,545	9861.45	568.32	5.76	2.6	68.8	1693.7	18
Pasco	6.1	37.7	17.2	16.8	12.5	195,114	3198.64	167.59	5.24	1.7	63.7	834.8	12
Piura	10.1	34.2	25	7.6	30.8	1,535,433	2694.29	420.4	15.6	1.9	66.8	992.6	81
Puno	10.8	37.9	20.4	9.3	14.1	904,267	2088.54	170.41	8.16	1.7	63.4	809.8	20
San Martín	11.3	36.5	19.9	14.9	25	639,533	3682.53	245.96	6.68	2.1	65.8	983.3	17
Tacna	10.5	38.7	34.4	9	26.7	303,701	4613.42	262.1	5.68	2.3	65.5	1259.9	26
Tumbes	11.4	40	27.6	14.5	25.8	191,850	4501.43	401.36	8.92	1.9	65.9	1142.6	8
Ucayali	7.4	37.7	22	17.5	30.7	416,932	4454.68	383.52	8.61	1.9	64.3	1203.1	18

Abbreviation: COVID-19, coronavirus disease 2019. * Confounding factors to multivariable regression models in 2020 included the mean age in deaths (years), mean monthly income (PEN), gender balance, and number of ICU beds. † SICOVID App. F500.2, SUSALUD, 2020 (accessed on 12 August 2022).

**Table 2 ijerph-19-11757-t002:** Prevalence rates of hypertension, overweight and obesity, smoking, and alcohol use and the incidence, mortality, and fatality rates due to COVID-19 in ≥15 year in 25 Peruvian regions in 2021.

Region	Hypertension Prevalence (%)	Overweight Prevalence (%)	Obesity Prevalence (%)	Prevalence of Tobacco Use (%)	Prevalence of Alcohol Use (%)	Population ≥ 15 Years in 2021	Incidence Rate (per 100,000 Habitants)	Mortality Rate (per 100,000 Habitants)	Case Fatality Rate (%)	Gender Balance in Deaths (Men/Women) *	Mean Age in Deaths (Years) *	No. of ICU Beds in 2021 *†	Vaccination Coverage in Individuals ≥ 18 Years (%) *
Amazonas	8.8	34.8	15.6	13.0	19.1	340,717	4084.33	203.39	4.98	1.7	66.4	20	68.0
Ancash	6.8	40.2	21.4	12.8	19.5	953,816	5092.07	405.21	7.96	1.7	67.3	49	88.4
Apurímac	6.0	35.1	13.7	12.3	29.6	351,074	5305.15	330.41	6.23	1.6	69.5	36	82.1
Arequipa	11.3	38.4	28.5	17.3	30.5	1,215,179	5560.17	451.29	8.12	1.6	65.9	53	84.5
Ayacucho	8.1	33.6	17.7	15.5	23.4	519,656	3685.52	254.59	6.91	1.6	68.2	20	70.6
Cajamarca	8.6	34.6	15.6	11.3	19.3	1,201,697	3387.38	212.62	6.28	1.7	68.0	54	75.4
Callao	11.6	38.7	30.8	21.4	25.9	877,161	6741.64	553.38	8.21	1.7	65.4	90	90.0
Cusco	10.5	37.6	18.1	13.1	26.7	1,111,868	4482.73	290.05	6.47	1.7	67.5	34	79.0
Huancavelica	8.8	31.1	10.4	13.1	13.3	330,566	2563.78	235.35	9.18	1.7	67.8	21	73.3
Huánuco	7.3	34.2	18.1	13.2	16.8	645,200	2445.44	255.27	10.44	1.9	67.1	31	69.7
Ica	8.2	35.1	35.0	14.4	29.4	700,394	4047.44	686.04	16.95	1.5	65.3	95	92.8
Junín	5.4	37.7	17.3	22.8	23.9	1,063,849	5538.47	437.66	7.9	1.8	66.2	64	81.2
La Libertad	8.3	37.1	28.2	10.8	23.1	1,544,977	3533.45	353.6	10.01	1.6	66.9	99	83.0
Lambayeque	9.9	39.2	23.8	8.2	31.8	1,050,982	2935.35	382.5	13.03	1.8	66.4	54	80.8
Lima	11.6	37.6	30.6	15.2	32.0	8,796,347	6206.02	536.17	8.64	1.7	65.5	662	88.8
Loreto	10.6	37.0	20.8	18.2	30.8	785,301	2301.03	181.71	7.9	1.5	64.4	36	64.5
Madre de Dios	10.6	39.6	31.9	27.9	26.2	129,465	3332.17	251.03	7.53	2.2	62.0	25	57.8
Moquegua	9.7	38.9	34.8	10.8	37.0	157,704	8552.1	410.26	4.8	1.8	66.9	24	83.7
Pasco	6.9	40.9	16.3	17.8	16.0	222,411	4259.23	323.72	7.6	1.4	64.0	29	81.4
Piura	10.0	35.8	27.0	10.0	35.9	1,521,251	3209.07	387.12	12.06	1.5	66.4	122	83.4
Puno	9.9	36.5	20.1	11.1	19.1	989,125	2248.05	273.47	12.16	2.1	64.5	40	57.5
San Martín	8.5	36.4	21.6	14.9	30.1	701,517	3178.97	206.55	6.5	1.6	66.2	29	73.6
Tacna	12.3	38.2	37.4	8.2	29.6	301,148	5434.54	393.16	7.23	2.0	63.8	35	75.2
Tumbes	10.2	38.9	31.0	13.4	24.8	181,769	5291.88	451.12	8.52	1.9	65.9	21	86.8
Ucayali	7.5	35.7	24.9	16.4	31.5	436,045	2579.09	342.63	13.28	1.8	64.5	28	64.3

Abbreviation: COVID-19, coronavirus disease 2019. * Confounding factors to multivariable regression models in 2021 included the mean age in the region (years), mean monthly income (PEN), gender balance, number of ICU beds, and COVID-19 vaccination coverage rates in 2021. † SICOVID App. F500.2, SUSALUD, 2021 (accessed on 12 August 2022).

**Table 3 ijerph-19-11757-t003:** Multiple regression analysis of the prevalence rates of obesity, smoking, and alcohol use with the incidence, mortality, and case fatality rates due to COVID-19 in 2020 *.

	No Adjusted Analysis					Full Adjusted Analysis				
Model	Coef.	SE	Beta (β)	*t*	*p*	Coef.	SE	Beta (β)	*t*	*p*
Crude incidence rate per 100,000 habitants										
Obesity prevalence	130.81	38.36	0.579	3.41	0.002	28.31	72.89	0.125	0.39	0.7
Smoking prevalence	144.96	82.19	0.345	1.76	0.091	−54.86	80.7	−0.13	−0.68	0.51
Prevalence of alcohol use	96.13	54.24	0.346	1.77	0.09	−50.73	55.3	−0.182	−0.92	0.37
Crude mortality rate per 100,000 habitants										
Obesity prevalence	15.14	2.46	0.787	6.13	0.0001	11.18	4.98	0.582	2.25	0.037
Prevalence of alcohol use	15.95	3.51	0.687	4.54	0.0001	8.09	4.47	0.348	1.81	0.087
Fatality case rate (%)										
Obesity prevalence	0.185	0.089	0.397	2.08	0.049	0.463	0.171	0.993	2.70	0.014

Abbreviation: COVID-19, coronavirus disease 2019; SE, standard error. * Model is adjusted for the following confounders: mean age in the region (cases or deaths), mean income and gender balance (cases or deaths), and the number of intensive care unit beds.

**Table 4 ijerph-19-11757-t004:** Multiple regression analysis of the prevalence rates of overweight and obesity with the COVID-19 mortality rate in 2021.

	Crude Mortality Rate per 100,000 Habitants (No Adjusted)					Crude Mortality Rate per 100,000 Habitants (Full Adjusted)				
	Coef.	SE	Beta (β)	*t*	*p*	Coef.	SE	Beta (β)	*t*	*p*
Overweight prevalence	17.92	10.57	0.333	1.69	0.104	8.84	11.07	0.164	0.80	0.434 *
Obesity prevalence	10.67	2.63	0.644	4.05	0.001	11.80	4.9	0.713	2.37	0.028 *
Overweight prevalence **	NA	NA	NA	NA	NA	−18.7	6.35	−0.349	−2.96	0.008
Obesity prevalence **	NA	NA	NA	NA	NA	0.52	3.94	0.031	0.13	0.895

Abbreviation: COVID-19, coronavirus disease 2019; SE, standard error; NA, not applicable. * Model is adjusted for the following confounders: mean age in the region (cases or deaths), mean income and gender balance (cases or deaths), and the number of intensive care unit beds. ** Model is adjusted for the confounders, including COVID-19 vaccination coverage rates in 2021.

## Data Availability

The data underlying the results presented in the study are available from: National Institute of Statistics and Informatics (INEI). Peruvian Population. Available online: https://www.inei.gob.pe/estadisticas/indice-tematico/population-estimates-andprojections/ (accessed on 3 June 2021). National Institute of Statistics and Informatics (INEI); Peru—National Demographic and Health Survey (ENDES) 2020. Noncommunicable and Communicable Diseases. 2020. Available online: https://proyectos.inei.gob.pe/endes/2020/SALUD/ENFERMEDADES_ENDES_2020.pdf (accessed on 20 January 2021). Noncommunicable and Communicable Diseases. 2020. Available online: Available online: https://proyectos.inei.gob.pe/endes/2021/SALUD/ENFERMEDADES_ENDES_2021.pdf (accessed on 22 July 2021). Peruvian Ministry of Health (MINSA). COVID-19 deaths. National System of Deaths (SINADEF). Available online: https://www.datosabiertos.gob.pe/dataset/fallecidos-porcovid-19-ministerio-de-salud-minsa (accessed on 3 June 2021). Superintendencia Nacional de Salud, Peru (SUSALUD). Daily Report on Form F500.2, App. for Centralized Management of the Availability of Hospitalization and ICU Beds at the National Level and of All Subsystems (Application F500.2); SUSALUD: Lima, Peru, 2020; Available online: http://portal.susalud.gob.pe/wp-content/uploads/archivo/registro-camas/2020/diciembre/31/20201231_turnonoche-7.-%20Reporte%20Ejecutivo%20-%20Comando%20COVID-19%20-%20F500.2%20(Camas)-%20ZC.pdf (accessed on 12 August 2022). Daily Report on Form F500.2, App. for Centralized Management of the Availability of Hospitalization and ICU Beds at the National Level and of All Subsystems (Application F500.2); SUSALUD: Lima, Peru, 2021; Available online: http://portal.susalud.gob.pe/wp-content/uploads/archivo/registro-camas/2021/diciembre/31/20211231_turnodia-1.-%20Reporte%20Ejecutivo%20-%20Comando%20COVID-19%20-%20F500.2%20(Camas)-%20ZC_v2.pdf (accessed on 12 August 2022). Peruvian Ministry of Health (MINSA). Single National Health Information Repository (REUNIS). COVID-19 vaccine in Peru. Available online: https://www.minsa.gob.pe/reunis/data/vacunas-covid19.asp (accessed on 31 December 2021).

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
