# Peer review of "COVID-19, Non-Communicable Diseases, and Behavioral Factors in the Peruvian Population ≥ 15 Years: An Ecological Study during the First and Second Year of the COVID-19 Pandemic"

_ijerph, 2022, doi:10.3390/ijerph191811757_

Round 1

Reviewer 1 Report

Well written manuscript and an interesting ecological study!

I just had a couple of comments/questions plus one correction .

1. I was curious to find out why you also include case fatality ratios (CFRs) measures in your study?

You show cumulative incidence and morality rates so I feel showing also CFRs is kind of redundant or was there any specific reason?

2. Why didn't you include population density as another confounder? Urbaness and closeness to infectious persons was a strong risk factor, at least in the beginning of the pandemic before stay at home orders. It seems like the mostly populated areas in Peru are Callao and Lima which also have the highest incidence and therefore mortality as well. Would it make sense to look at urban regions (Callao and Lima maybe?) vs rural regions?

Line 39: I would correct the sentence to: "hypertension and obesity are significant risk factors for severe illness and death." In order to contract COVID-19 you have to be exposed to an infectious person first, independent of any risk factor but the disease severity once infected might be different.

Author Response

Response-to-reviewers: Manuscript ID ijerph-1893232

We thank the Reviewer for your comments and constructive criticism, we believe that the quality of our manuscript has been significantly improved. We have revised our paper in a point-by-point manner.

Reviewer 1

Well written manuscript and an interesting ecological study! I just had a couple of comments/questions plus one correction.

Response: Thank you for your comments.

Comment 1. I was curious to find out why you also include case fatality ratios (CFRs) measures in your study? You show cumulative incidence and morality rates so I feel showing also CFRs is kind of redundant or was there any specific reason?

Response 1: Thank you for your comments. The cumulative incidence and mortality rates were estimates for overall population (per 100,000 habitants), while CFRs are estimates of number of deaths by number of cases. They are different estimates. In addition, we looked for all possible epidemiological indicators to explain the association with the study variables.

Comments 2: Why didn't you include population density as another confounder? Urbaness and closeness to infectious persons was a strong risk factor, at least in the beginning of the pandemic before stay at home orders. It seems like the mostly populated areas in Peru are Callao and Lima which also have the highest incidence and therefore mortality as well. Would it make sense to look at urban regions (Callao and Lima maybe?) vs rural regions?

Response 2: Thank you for your comments. The cumulative incidence and mortality rates were estimates for overall population (per 100,000 habitants), in these calculations the total population of each region was already used as the denominator. ENDES and Peruvian population data are not available by urban or rural. The analyzes are for the general population.

Comments 3: Line 39: I would correct the sentence to: "hypertension and obesity are significant risk factors for severe illness and death." In order to contract COVID-19 you have to be exposed to an infectious person first, independent of any risk factor but the disease severity once infected might be different.

Response 3: Thank you for your comments. The sentence was corrected.

Reviewer 2 Report

The topic is up-to-date and the study - well-designed, but some improprieties need to be corrected.

1.     Page 3, lines 119-122 – please complete - the rates are crude or standardized?

2.     Page 4, lines 149-153 - please describe precisely in which region, which rate had the highest value and present values in the text.

3.     Page 5 - please update the page numbering.

4.     Page 5 – Table 1 - lack of units of rates - [per …].

5.     Page 5-6 – Figure 2 and Figure 3 - Linear regression analysis is not possible in the case of the Spearman correlation. The figures contain the equations of linear functions, which should be removed.

6.     Page 6-7 – please change from “p=0.000” to “p<0.0001” in the text and in Table 2.

7.     Page 6-8 – Table 2 and Table 3 - please present the results of the "no adjusted" and "full adjusted" models next to each other.

8.     Page 7, line 186 – “r=0.45; p=0.024” – inconsistency of the result in the text and in the figure.

Author Response

Response-to-reviewers: Manuscript ID ijerph-1893232

We thank the Reviewer for your comments and constructive criticism, we believe that the quality of our manuscript has been significantly improved. We have revised our paper in a point-by-point manner.

Reviewer 2

The topic is up-to-date and the study - well-designed, but some improprieties need to be corrected.

Comment 1. Page 3, lines 119-122 – please complete - the rates are crude or standardized?

Response 1: Thank you for your comments. The sentence was corrected.

Comment 2. Page 4, lines 149-153 - please describe precisely in which region, which rate had the highest value and present values in the text.

Response 2: Thank you for your comments. The sentence was corrected.

Comment 3. Page 5 - please update the page numbering.

Response 3: Thank you for your comments. The sentence was corrected.

Comment 4. Page 5 – Table 1 - lack of units of rates - [per …].

Response 4: Thank you for your comments. The Tables 1 was corrected. We also include a table 2 with the updated data from ENDES 2022. Our results have not changed.

Comment 5. Page 5-6 – Figure 2 and Figure 3 - Linear regression analysis is not possible in the case of the Spearman correlation. The figures contain the equations of linear functions, which should be removed.

Response 5: Thank you for your comments. In Figures 2 and 3, we include 2 types of analysis, linear regression models and Spearman correlation between the prevalence rates of hypertension, overweight, obesity, smoking, and alcohol use and the incidence rate, mortality rate and case fatality rate due to COVID-19 in 2020 and 2021. We consider that it is not necessary to eliminate the linear regression models of each Figure. This may help the reader to understand the differences between the regression model and the correlation analysis, as shown in other studies.

Comment 6. Page 6-7 – please change from “p=0.000” to “p<0.0001” in the text and in Table 2.

Response 6: Thank you for your comments. The sentence was corrected.

Comment 7.     Page 6-8 – Table 2 and Table 3 - please present the results of the "no adjusted" and "full adjusted" models next to each other.

Response 7: Thank you for your comments. The Tables 3 and 4 was corrected.

Comment 8. Page 7, line 186 – “r=0.45; p=0.024” – inconsistency of the result in the text and in the figure.

Response 8: Thank you for your comments. The sentence was corrected.